# STRIDE: TRAINING DATA ATTRIBUTION CAN BE ESTIMATED IN ACTIVATION SPACE

.   **Abir Harrasse**[1,2]   **Rishit Dagli**[1]   **Amirali Abdullah**[2,3]   **Zhijing Jin**[1,4,5]
[1]Jinesis AI Lab, University of Toronto & Vector Institute   [2]Martian AI   [3]Thoughtworks
[4]EuroSafeAI   [5]Max Planck Institute for Intelligent Systems

## ABSTRACT

Understanding which training examples drive specific model behaviors is central to debugging failures, investigating safety issues, and auditing deployed systems. However, existing attribution methods operate in parameter space, where costs grow rapidly with model size. Approximations enable scaling, but introduce overhead that limits low-latency and scalable deployment. STRIDE is a scalable framework that estimates influence directly in activation space, bypassing explicit parameter interactions. STRIDE learns low-rank steering operators that approximate the effect of retraining on data subsets by shifting internal representations. We then recover per-example influence scores by solving a regularized regression problem that decomposes these subset-level shifts. Experiments show that STRIDE accurately identifies influential examples and detects data leakage, outperforming prior methods while being orders of magnitude faster and scalable.

## 1 INTRODUCTION

Modern neural networks are trained on massive, heterogeneous corpora that preclude direct inspection (Brown et al., 2020; Gao et al., 2020; Weber et al., 2024). When failures arise—such as spurious correlations (Du et al., 2021), unsafe outputs (Gehman et al., 2020; Perez et al., 2022), data leakage (Carlini et al., 2021), or copyright violations (Sag, 2023)—it is often unclear which training examples are responsible. Training Data Attribution (TDA) seeks to quantify the influence of individual training points on model behavior (Koh & Liang, 2017), but most existing approaches were developed for pre-LLM regimes where retraining and parameter-space influence estimation remained tractable.

As the societal stakes of LLM failures grow (Bommasani et al., 2022; Weidinger et al., 2021), scalable attribution becomes essential for interpretability, fairness, and data provenance. However, leave-one-out retraining is infeasible, and practical approximations rely on parameter-space gradients or curvature estimates that do not scale to modern models (Pruthi et al., 2020).

We propose estimating training data influence directly in activation space. On smaller models and datasets, this approach matches or exceeds parameter-space baselines while reducing compute and memory costs by orders of magnitude. Our contributions are:

- We propose STRIDE, the first framework to estimate training data influence entirely in activation space, avoiding parameter-space scaling issues.
- We introduce a gradient-free, two-stage method that models subset-level activation shifts via low-rank operators and recovers per-example influence with sparse regression.
- We show that STRIDE matches or outperforms baselines on standard benchmarks, detecting data leakage and mislabeled examples with much lower compute.

## 2 RELATED WORK

**Training Data Attribution (TDA).**   TDA quantifies the contribution of training examples to model predictions. Exact influence can be measured via leave-one-out retraining and case-deletion analysis (Cook, 1977; Koh & Liang, 2017), but these approaches are computationally infeasible for

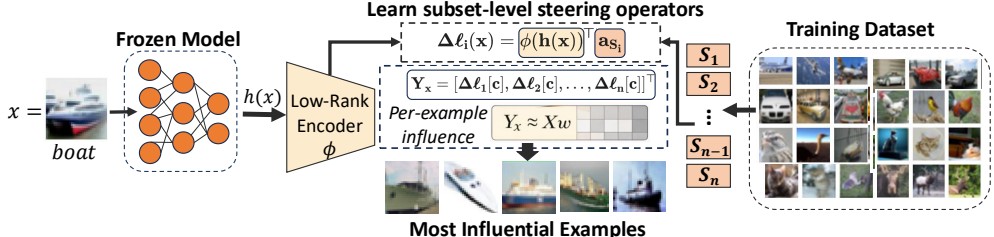

Figure 1: **STRIDE: Activation-Space Influence Estimation**. STRIDE uses a frozen pretrained model to extract features $h(x)$ from a test input $x$ with predicted label $c$ (boat). A low-rank encoder $\phi$ learns input-dependent steering operators paired with subset-specific matrices $a_{S_i}$ to estimate per-subset influence. Subset-level influence scores $Y_x$ are then transformed to per-example influences $w$ via regression over the subset membership matrix $X$. The LDS constraint (dashed box) regularizes $w$ to respect subset structure, enabling accurate identification of influential training examples.

modern models. Scalable approximations fall into three main families. Similarity-based heuristics (e.g., $k$-NN in representation space) provide efficient (Akyürek et al., 2022) but correlational proxies (Papernot & McDaniel, 2018; Yeh et al., 2018; Rajani et al., 2020). Gradient-based methods, beginning with influence functions (Koh & Liang, 2017), linearize retraining dynamics using parameter derivatives, with later extensions leveraging training trajectories (Pruthi et al., 2020; Park et al., 2023) and scaling techniques such as preconditioning and sketching (Wang et al., 2025; Choe et al., 2024). Orthogonally, data valuation methods such as Data Shapley assign influence via cooperative game-theoretic marginal contributions (Ghorbani & Zou, 2019; Jia et al., 2019), but require costly retraining approximations.

While gradient-based methods inherit parameter-space costs and representation-based approaches like AirRep (Sun et al., 2025) rely on similarity in a learned embedding space, STRIDE operates via learned steering operators that causally intervene on internal activations to approximate retraining effects.

**Activation-Space Steering.** Our work draws inspiration from activation-space steering, where internal representations are modulated to control model behavior. Techniques like Activation Addition (Turner et al., 2024), classifier gradient steering (Oozeer et al., 2025), SAE steering (O'Brien et al., 2024) and Contrastive Activation Addition (Panickssery et al., 2024) construct steering vectors to elicit specific capabilities or suppress toxicity. While prior steering work focuses on model control or concept editing (Dunefsky & Cohan, 2025), we reframe steering as a mechanism for attribution. By learning operators that simulate the effect of removing data subsets, STRIDE leverages the efficiency of activation engineering to solve the training data attribution problem without ever touching model parameters.

## 3 METHOD

Let $f_\theta : \mathcal{X} \to \mathbb{R}^C$ be a frozen classifier and $h(x) \in \mathbb{R}^d$ an internal-layer activation. Given training data $\mathcal{D} = (x_i, y_i)_{i=1}^N$ and test input $z$, STRIDE estimates per-example influence scores $w \in \mathbb{R}^N$ indicating how each training point contributes to $z$, without retraining.

STRIDE assumes removing a small subset induces an approximately additive, locally linear perturbation in activation space (Fig. 1).

**Activation-Space Influence Model.** For subset $S$, we model the logit shift as $\Delta\ell_S(h) = a_S^\top \phi(h)$, with $\phi : \mathbb{R}^d \to \mathbb{R}^r$ a shared encoder and $a_S \in \mathbb{R}^{r \times C}$ a subset-specific steering matrix. Low-rank factorization constrains influence to a few latent directions, enabling efficiency.

**Subset Factorization.** We partition the training set into subsets $\{S_k\}_{k=1}^K$, each with a steering operator $a_k$. For a test input $z$ with predicted class $c$, subset responses are $Y_z^{(k)} = (\phi(h(z))^\top a_k)_c$,

---

**Algorithm 1** STRIDE: Activation-Space Influence Estimation

---

1: **Input:** Frozen model $f_\theta$, training data $\mathcal{D}$, subsets $\{S_k\}_{k=1}^K$, test input $z$
2: **Output:** Influence scores $w^* \in \mathbb{R}^N$
3: **Stage 1: Learn operators**
4: **for** $k = 1$ to $K$ **do**
5:     Initialize $\phi$ and $a_k$; optimize $\mathcal{L}$fid $+ \lambda_{\text{stab}}\mathcal{L}$stab $+ \lambda\text{lin}\mathcal{L}_{\text{lin}}$
6:     on mini-batches $B_S \subset S_k$, $B_R \subset \mathcal{D} \setminus S_k$
7: **end for**
8: **Stage 2: Decompose influence**
9: $v_z = \phi(h(z))$, $Y_z^{(k)} = (v_z^\top a_k)_c$, $Y_z = [Y_z^{(1)}, \ldots, Y_z^{(K)}]^T$
10: Solve $w^* = \arg\min_w \frac{1}{2}||Y_z - Xw||_2^2 + \alpha||w||_1$
11: **return** $w^*$

---

stacked into $Y_z \in \mathbb{R}^K$. Let $X \in \{0, 1\}^{K \times N}$ denote subset membership. Assuming linear superposition, $Y_z = Xw + \varepsilon$, where $w$ is sparse and recovered via $\ell_1$-regularized least squares, and $\varepsilon$ captures residual nonlinearities and interactions.

**Learning Objectives.** Operators are optimized to (i) match retraining behavior on their subset, (ii) remain inactive on held-out examples, and (iii) compose linearly. Fidelity, stability, and linearity losses are in Appendix A. Algorithm 1 summarizes the procedure.

## 4 EXPERIMENTS AND RESULTS

We evaluate STRIDE on MNIST (LeCun et al., 2002) and CIFAR-10 (Krizhevsky et al., 2009) using frozen pretrained models (MLP and CNN). Our primary goal is to validate that STRIDE achieves high fidelity in influence estimation while dramatically reducing computational costs compared to parameter-space methods. We compare against TracIn (Pruthi et al., 2020) and EKFAC (Grosse et al., 2023). Full experimental details, hyperparameters, and extensive additional results are provided in Appendix D.

**Fidelity of Influence Estimation.** We measure the quality of influence scores using the Linear Datamodeling Score (LDS) (Park et al., 2023). Table 1 presents the Spearman and Pearson correlations of our estimated influence against ground-truth LDS for both MNIST and CIFAR-10. STRIDE achieves fidelity that is competitive with or superior to parameter-space baselines. On CIFAR-10, we achieve a Spearman correlation of $0.32$, outperforming EKFAC (Grosse et al., 2023) $(0.29)$ and TracIn (Pruthi et al., 2020) $(0.28)$.

**Efficiency and Batch Inference.** Crucially, STRIDE is orders of magnitude faster during inference $(1.7s$ vs $27s)$, as shown in Table 1. Unlike EKFAC (Grosse et al., 2023), which requires careful batching to avoid OOM errors due to large curvature matrices, STRIDE operates purely on activation vectors. This allows for trivial batching and massive parallelization, making it uniquely suitable for large-scale influence estimation where throughput is critical.

| MNIST | | | | | CIFAR-10 | | | | |
|---|---|---|---|---|---|---|---|---|---|
| Method | Spearman (↑) | Pearson (↑) | Setup (s) (↓) | Inf (s) (↓) | Method | Spearman (↑) | Pearson (↑) | Setup (s) (↓) | Inf (s) (↓) |
| TracIn (Pruthi et al., 2020) | $0.103 \pm 0.18$ | $\mathbf{0.107 \pm 0.13}$ | – | $\underline{22.35}$ | TracIn (Pruthi et al., 2020) | $0.276 \pm 0.25$ | $0.655 \pm 0.21$ | – | $\underline{28.76}$ |
| EKFAC (Grosse et al., 2023) | $0.105 \pm 0.19$ | $0.106 \pm 0.13$ | $\mathbf{17.18}$ | $28.10$ | EKFAC (Grosse et al., 2023) | $\underline{0.288 \pm 0.21}$ | $\mathbf{0.738 \pm 0.17}$ | $\mathbf{23.28}$ | $27.10$ |
| STRIDE | $\mathbf{0.116 \pm 0.06}$ | $0.057 \pm 0.11$ | $77.18$ | $\mathbf{2.21}$ | STRIDE | $\mathbf{0.319 \pm 0.05}$ | $\underline{0.663 \pm 0.08}$ | $110.38$ | $\mathbf{1.72}$ |

Table 1: **Fidelity and Efficiency Comparison.** STRIDE matches or exceeds baselines in correlation (Spearman/Pearson) while significantly reducing amortized inference time.

**Computational Scalability.** The key advantage of STRIDE is efficiency. By avoiding gradient computations and Hessian approximations, STRIDE scales linearly with the representation dimension rather than parameter count. Figure 2 in Appendix D.1 demonstrates that STRIDE maintains low inference latency even as model size increases, whereas EKFAC (Grosse et al., 2023) runs out of memory (OOM) for larger models. Further, we analyze the computational efficiency of STRIDE compared to parameter-space influence methods.

*Proposition* 1 (Complexity). Let $N$ be the dataset size, $P$ the number of model parameters, $L$ the number of layers, $d$ the activation dimension, $r$ the steering rank, $K$ the number of subsets, and $s$ the average sparsity (non-zeros per row in the membership matrix). Assume $r \approx d \ll P$, $K \ll N$, and $s \ll K$. The asymptotic complexities for Space (memory), Setup Time (pre-computation), and Inference Time (per test point) are:

| METHOD | SPACE | SETUP TIME | INFERENCE TIME |
|--------|-------|-----------|----------------|
| EKFAC | $\mathcal{O}(Ld^2)$ | $\mathcal{O}(N_{\text{FISHER}}Ld^3)$ | $\mathcal{O}(NLd^3)$ |
| STRIDE | $\mathcal{O}(Ns + Kd^2)$ | $\mathcal{O}(Ns + TKd^2)$ | $\mathcal{O}(T_{\text{LASSO}}Ks + Kd^2)$ |

where $T$ is the number of operator optimization iterations and $T_{\text{lasso}}$ is the number of Lasso iterations. For EKFAC, inference requires computing gradients for all $N$ training examples on-the-fly. For typical transformers where $P = \Theta(Ld^2)$, STRIDE with sparse membership ($s \ll K$) achieves substantially lower complexity: setup time is reduced from $\mathcal{O}(N_{\text{Fisher}}Ld^3)$ to $\mathcal{O}(TKd^2)$, and inference is reduced from $\mathcal{O}(NLd^3)$ to $\mathcal{O}(T_{\text{lasso}}Ks + Kd^2)$, effectively converting dependence on dataset size $N$ and cubic scaling in dimension to dependence on subset count $K$ and sparsity $s$.

See Appendix F for a derivation.

**STRIDE Setup Time Analysis.** STRIDE's setup time includes an $O(Ns)$ term for sketching the membership matrix. However, this operation can exploit matrix sparsity and is embarrassingly parallel (Appendix F.3), unlike EKFAC's $O(N_{\text{FISHER}}Ld^3)$ which requires sequential gradient computations through the full model. Moreover, setup cost is amortized across inference: STRIDE's 10-16× faster inference (1.7s vs 27s on CIFAR-10) means total cost becomes favorable after a small number of test queries.

**Downstream Applications.** We further validate STRIDE on practical tasks:

- **Mislabeled Data Detection**: STRIDE identifies flipped labels with high precision (92% on MNIST, 63% on CIFAR-10), outperforming baselines (Appendix D.4).
- **Data Leakage Detection**: We successfully detect identical training examples with perfect accuracy in toy settings, ranking leaked examples at the top (Appendix D.4).
- **Counterfactual Policy**: Removing top influential examples identified by STRIDE effectively drops model probability for test samples, confirming the causal validity of our scores (Appendix D.4).

## 5 DISCUSSION AND CONCLUSION

We presented STRIDE, a method that shifts influence estimation from parameter space to activation space. By learning steering operators that linearly approximate the effect of training data subsets, we achieve fidelity comparable to state-of-the-art methods like EKFAC and TracIn while reducing computational cost by orders of magnitude. This efficiency enables influence estimation to scale to modern foundation models where traditional gradient-based approaches are computationally prohibitive.

**Limitations.** Our approach relies on the assumption that influence can be modeled as a linear transformation in the representation space. While our ablations confirm that linearity constraints improve generalization, this approximation may overlook highly non-linear interactions between training points and model predictions. Additionally, our method currently aggregates influence at the subset level. While this improves signal-to-noise ratio and efficiency, it assumes that data within a random subset contributes somewhat homogeneously; extremely outlier-heavy subsets might require finer-grained analysis. Finally, like most training data attribution methods, we operate on frozen representations, which is standard for influence functions and does not capture full end-to-end dynamics, although our formulation could in principle track influence throughout training.

**Future Work.** The dramatic efficiency of STRIDE opens several avenues. First, extending this framework to Large Language Models (LLMs) is a natural next step, as the computational gap between $O(NK)$ and $O(NP)$ becomes even more critical for billion-parameter transformers. Second,

investigating dynamic subset selection or hierarchical steering could allow for adaptive resolution, zooming in on influential data regions without processing the entire dataset at the instance level. Finally, exploring non-linear steering operators could further bridge the fidelity gap in highly complex feature spaces.

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

APPENDIX

## A  TRAINING OBJECTIVES

We jointly optimize the feature map $\phi$ and prediction matrices $\{W_1, \ldots, W_K\}$ using a combined loss with three components, each targeting a different aspect of influence estimation.

### A.1  FIDELITY LOSS: $\mathcal{L}_{\text{fid}}^{(k)}$

The fidelity loss ensures that the learned operator $g_k$ accurately predicts how removing subset $\mathcal{S}_k$ affects model predictions. For each subset, we measure cross-entropy between true labels and predictions after applying the operator:

$$\mathcal{L}_{\text{fid}}^{(k)} = \frac{1}{|\mathcal{S}_k|} \sum_{(x,y) \in \mathcal{S}_k} H(y, \sigma(z(x) + W_k^\top \phi(h(x)))), \tag{1}$$

where $z(x) = f_\theta(x)$ are the original model logits, $H(y, \cdot)$ is cross-entropy, and $\sigma$ is softmax. This loss encourages $g_k$ to shift predictions toward the distribution expected if $\mathcal{S}_k$ had been removed during training.

### A.2  STABILITY LOSS: $\mathcal{L}_{\text{stab}}^{(k)}$

Operators must remain localized, affecting only the subset they represent. We penalize prediction changes on out-of-distribution data by measuring KL divergence to the original predictions:

$$\mathcal{L}_{\text{stab}}^{(k)} = \frac{1}{|\mathcal{R}_k|} \sum_{x \in \mathcal{R}_k} D_{\text{KL}}(\sigma(z(x)) \| \sigma(z(x) + W_k^\top \phi(h(x)))), \tag{2}$$

where $\mathcal{R}_k$ is a random subset sampled from $D \backslash \mathcal{S}_k$. This prevents operators from generating spurious effects on unrelated examples.

### A.3  LINEAR AGGREGATION LOSS: $\mathcal{L}_{\text{lin}}^{(k)}$

Influence must decompose linearly across training examples: the aggregate effect on subset $\mathcal{S}_k$ should equal the sum of individual example effects. We define subset-level aggregates:

$$y_k = \frac{1}{|\mathcal{S}_k|} \sum_{x \in \mathcal{S}_k} (W_k^\top \phi(h(x)))_{\text{max}}, \tag{3}$$

where $(\cdot)_{\text{max}}$ extracts the response for the most-activated class. We then enforce that per-example responses aggregate to this value via ridge regression in a sketched space:

$$\mathcal{L}_{\text{lin}}^{(k)} = \left\| \tilde{X}_k \left( (\tilde{X}_k^\top \tilde{X}_k + \gamma I)^{-1} \tilde{X}_k^\top y_k \right) - y_k \right\|_2^2, \tag{4}$$

where $\tilde{X}_k = X_k R$ applies Johnson-Lindenstrauss projection with random matrix $R \in \mathbb{R}^{N \times 64}$ to the subset membership matrix $X_k \in \{0, 1\}^{|\mathcal{S}_k| \times N}$.

### A.4  COMBINED OBJECTIVE

The total training loss combines all three terms:

$$\mathcal{L} = \sum_{k=1}^{K} \left( \mathcal{L}_{\text{fid}}^{(k)} + \lambda_{\text{stab}} \mathcal{L}_{\text{stab}}^{(k)} + \lambda_{\text{lin}} \mathcal{L}_{\text{lin}}^{(k)} \right), \tag{5}$$

where $\lambda_{\text{stab}}$ and $\lambda_{\text{lin}}$ are non-negative regularization weights. This formulation ensures that operators are faithful to retraining effects (fidelity), localized to their subsets (stability), and decomposable into per-example influences (linearity).

| MNIST (MLP) | CIFAR-10 (CNN) |
|---|---|
| Input: $28 \times 28$ Flattened (784) | Input: $3 \times 32 \times 32$ |
| Linear(784, 512) + ReLU | Conv(3, 32, 3, pad=1) + BN + ReLU + MaxPool(2) |
| Linear(512, 256) + ReLU | Conv(32, 64, 3, pad=1) + BN + ReLU + MaxPool(2) |
| (Feature Extraction Layer) | AdaptiveAvgPool((4,4)) + Flatten |
| Linear(256, 10) | Linear(1024, 512) + ReLU + Dropout(0.5) |
| | (Feature Extraction Layer) |
| | Linear(512, 10) |

Table 2: **Summary of Model Architectures.** Layer-wise architectures for the MLP (MNIST) and CNN (CIFAR-10) models used in our experiments, including the designated layers for feature extraction.

| Parameter | MNIST | CIFAR-10 |
|---|---|---|
| Optimizer | Adam | SGD |
| Learning Rate | 0.001 | 0.01 |
| Batch Size | 64 | 64 |
| Weight Decay | 0 | $10^{-4}$ |
| Momentum | - | 0.9 |
| Epochs | 20 | 40 |

Table 3: **Base Model Training Hyperparameters.** Optimization settings (learning rate, optimizer, batch size, etc.) used to train the frozen base models prior to influence estimation.

## B    EXPERIMENTAL SETUP AND HYPERPARAMETERS

We provide additional experimental details to ensure reproducibility. We distinguish between the setup for our main fidelity/downstream experiments and the synthetic scalability benchmarks.

### B.1    MODEL ARCHITECTURES

We evaluate STRIDE on two distinct architectures representing common paradigms: a fully connected Multilayer Perceptron (MLP) for MNIST and a Convolutional Neural Network (CNN) for CIFAR-10. Table 2 details the layer specifications for both models. For the MLP, we extract features from the penultimate layer (dimension 256). For the CNN, we extract features from the output of the fully connected layer following the convolutional blocks (dimension 512).

### B.2    TRAINING HYPERPARAMETERS

**Base Models.**    We trained our base models using a standard grid search to achieve competitive performance. The final hyperparameters selected are listed in Table 3. The MLP achieves $98.53\%$ test accuracy, and the CNN achieves $81.23\%$ test accuracy.

**Steering Operators.**    For STRIDE, we freeze the base model and learn low-rank steering operators. Table 4 lists the optimal hyperparameters used for the results reported in this paper. These were selected based on stability and fidelity on a validation set. We use a subset size of $|\mathcal{S}_k| = 100$ and split the training data into $K = 50$ subsets for analysis.

### B.3    SCALABILITY ANALYSIS

For the scalability benchmarking reported in Appendix D.1, we employ a controlled synthetic environment to isolate the effect of model size on computational cost.

**Synthetic Data.**    We generate a synthetic dataset of 5,000 training examples and 100 test examples with input dimensions $3 \times 32 \times 32$ (matching CIFAR-10 structure). This ensures that data loading overheads are consistent across runs and independent of disk I/O.

| Parameter | MNIST | CIFAR-10 |
|---|---|---|
| Rank ($r$) | 32 | 64 |
| Learning Rate ($\eta$) | 0.005 | 0.001 |
| Iterations | 250 | 200 |
| LDS Regularization ($\lambda_{\text{lin}}$) | 10.0 | 1.0 |
| Stability Regularization ($\lambda_{\text{stab}}$) | 1.0 | 1.0 |

Table 4: **Steering Operator Hyperparameters.** Optimal hyperparameters for learning steering vectors, selected based on validation set stability and fidelity metrics.

**Scalable Architectures.** We utilize a parametric `ScalableCNN` architecture that allows us to vary depth and width systematically. The network consists of a customizable number of convolutional blocks followed by a linear head. We sweep depths from 4 to 24 layers and widths from 64 to 768 channels to span a model size range from 15M to over 1B parameters. This allows us to stress-test the memory and runtime limits of each attribution method without architectural confounds.

## C  METRICS

Let $\hat{w} \in \mathbb{R}^N$ denote STRIDE per-example influence scores for test input $z$.

### C.1  SPEARMAN CORRELATION WITH LDS

Let $\text{LDS} \in \mathbb{R}^N$ be the linearized influence score per training example. Spearman correlation is

$$\rho(\hat{w}, \text{LDS}) = \frac{\text{cov}(\text{rank}(\hat{w}), \text{rank}(\text{LDS}))}{\sigma(\text{rank}(\hat{w}))\sigma(\text{rank}(\text{LDS}))}.$$

### C.2  FLIPPED-LABEL DETECTION

Given the set $F$ of flipped training labels, top-$M$ predicted examples by $\hat{w}$:

$$\text{Precision} = \frac{|\text{top}_M(\hat{w}) \cap F|}{M}.$$

### C.3  DATA LEAKAGE DETECTION

Given a leakage set $\mathcal{L}$ and top-$k$ examples inferred by $\hat{w}$:

$$\text{Leakage Rate} = \frac{|\text{top}_k(\hat{w}) \cap \mathcal{L}|}{k}.$$

### C.4  COUNTERFACTUAL EVALUATION

Let $X_k$ be the top-$k$ examples by $\hat{w}$. Let $f_\theta^{-X_k}$ denote the model hypothetically retrained without $X_k$. Then probability drop for class $c$ is

$$\Delta p_k(z) = \sigma(f_\theta(z))_c - \sigma(f_\theta^{-X_k}(z))_c.$$

## D  ADDITIONAL RESULTS

In this section, we provide detailed experimental setups, extended quantitative results, and scalability analysis.

### D.1  SCALABILITY ANALYSIS

We compare the wall-clock time and memory usage of STRIDE vs. EKFAC (Grosse et al., 2023) across varying model sizes (depth and width) on our synthetic dataset of 5,000 examples. Figure 2 visualizes the Setup Time (pre-computation) and Inference Time (attribution per test point).

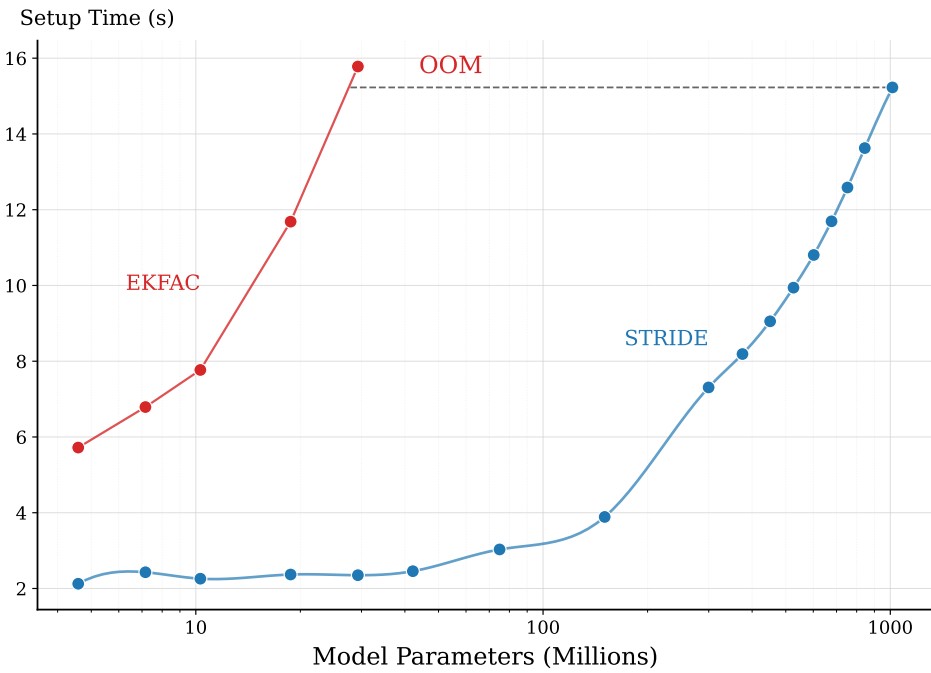

Figure 2: **Scalability Comparison.** Setup times for STRIDE vs. EKFAC (Grosse et al., 2023). STRIDE remains efficient and stable, while EKFAC (Grosse et al., 2023) errors out (OOM) for larger models.

| Depth | Width | Params | EKFAC (Grosse et al., 2023) Setup (s) | STRIDE Setup (s) |
|-------|-------|--------|----------------------------------------|------------------|
| 4 | 64 | 262K | 6.43 | 3.27 |
| 6 | 128 | 4.6M | 5.72 | 2.13 |
| 6 | 160 | 7.2M | 6.79 | 2.43 |
| 6 | 192 | 10.3M | 7.77 | 2.26 |
| 8 | 128 | 18.8M | 11.68 | 2.37 |
| 8 | 160 | 29.3M | 15.78 | 2.35 |
| 8 | 192 | 42.2M | OOM ✗ | 2.46 |
| 8 | 256 | 75.0M | OOM ✗ | 3.03 |
| 10 | 256 | 150.5M | OOM ✗ | 3.89 |
| 12 | 512 | 299.7M | OOM ✗ | 7.31 |
| 14 | 512 | 375.2M | OOM ✗ | 8.19 |
| 16 | 512 | 450.7M | OOM ✗ | 9.05 |
| 18 | 512 | 526.3M | OOM ✗ | 9.94 |
| 20 | 512 | 601.8M | OOM ✗ | 10.80 |
| 22 | 512 | 677.3M | OOM ✗ | 11.69 |
| 24 | 512 | 752.8M | OOM ✗ | 12.59 |
| 14 | 768 | 844.2M | OOM ✗ | 13.63 |
| 16 | 768 | 1.0B | OOM ✗ | 15.23 |

Table 5: **Scalability Comparison (Full Data).** Wall-clock time (seconds) for Setup phase across varying model depths and widths. 'OOM' indicates Out-of-Memory failures, highlighting the scalability limitations of EKFAC (Grosse et al., 2023) on larger models.

Table 5 details the raw data used for the scalability plot. Note that EKFAC (Grosse et al., 2023) fails (OOM) for models with depth $\geq 8$ in our setup, whereas STRIDE scales well.

| Method | Layer | Spearman (↑) |
|---|---|---|
| No LDS Loss | Penultimate | **0.149 ± 0.09** |
| Layer Sens. | layer1 | -0.012 ± 0.12 |

| Method | Layer | Spearman (↑) |
|---|---|---|
| No LDS Loss | Penultimate | 0.248 ± 0.10 |
| Layer Sens. | conv | 0.223 ± 0.06 |

Table 6: **Ablation Study Results (MNIST).** Comparison of influence estimation fidelity (LDS Spearman correlation) on MNIST when removing key components like the LDS regularization term or changing the target layer.

Table 7: **Ablation Study Results (CIFAR-10).** Evaluation of component importance on CIFAR-10, highlighting the criticality of the stability and linearity constraints for robust influence estimation in deeper networks.

| Method | k=50 (↑) | k=100 (↑) | k=500 (↑) | k=1000 (↑) |
|---|---|---|---|---|
| ClassRandom | 0.001 ± 0.00 | 0.013 ± 0.01 | 0.352 ± 0.06 | 0.538 ± 0.08 |
| EKFAC (Grosse et al., 2023) | 0.002 ± 0.01 | **0.016 ± 0.02** | 0.438 ± 0.12 | 0.737 ± 0.11 |
| TracIn (Pruthi et al., 2020) | **0.003 ± 0.01** | 0.012 ± 0.02 | 0.424 ± 0.13 | 0.728 ± 0.13 |
| STRIDE | 0.001 ± 0.00 | 0.015 ± 0.03 | **0.471 ± 0.12** | **0.811 ± 0.10** |

Table 8: **Counterfactual Evaluation on MNIST.** Influence effectiveness measured by the probability drop of the true class after retraining on the top-$k$ most helpful examples identified by each method. Higher probability drop indicates better influence estimation.

## D.2 INFLUENCE ESTIMATION FIDELITY

We compare the Spearman and Pearson correlations of influence scores with ground-truth LDS scores in Table 1 (Main Text). STRIDE achieves consistently high correlation, indicating strong agreement with retraining-based influence.

## D.3 ABLATION STUDIES

We perform ablations to verify the importance of the LDS-specific regularization and layer selection. 'No LDS Loss' removes the stability/linearity constraints. 'Layer Sensitivity' tests deriving influence from earlier layers. Results in Tables 6 and 7 show mixed results on MNIST but clear trends on CIFAR-10. On the simpler MNIST dataset, unregularized regression 'No LDS Loss') performs slightly better, likely because the problem is well-conditioned. However, on the more complex CIFAR-10 task, our full regularized objective ('STRIDE') significantly outperforms the ablations, confirming that stability and linearity constraints are critical for generalization in deep representation spaces.

## D.4 DOWNSTREAM TASKS

**Counterfactual Evaluation.** We evaluate the influence of training data by removing the top-$k$ most helpful examples (as identified by each method) and retraining the model. A larger drop in the true class probability indicates that the method has correctly identified the data responsible for the prediction. Results in Tables 8 and 9 show a clear trend: while parameter-space methods like EKFAC (Grosse et al., 2023) and TracIn (Pruthi et al., 2020) perform similarly to STRIDE for small $k$ ($k = 50$), STRIDE dominates as the cohort size increases. On CIFAR-10 with $k = 1000$, removing our identified examples causes a probability drop of $0.915$, significantly higher than EKFAC (Grosse et al., 2023)'s $0.846$. This suggests that STRIDE's activation-space approach better captures the *cumulative* effect of large data groups, whereas curvature approximations in EKFAC (Grosse et al., 2023) may degrade when summing over many data points.

**Mislabeled Data Detection.** We simulate label noise by flipping the labels of a random $10\%$ subset of the training data. A reliable attribution method should assign high *negative* influence to these detrimental examples, allowing them to be flagged. STRIDE demonstrates superior detection capabilities, achieving a Precision/Recall of $0.92$ on MNIST and $0.63$ on CIFAR-10 (Tables 10 and 11), substantially outperforming EKFAC (Grosse et al., 2023) ($0.85$ and $0.51$, respectively). We hypothesize that mislabeled examples induce distinct, high-magnitude distortions in the representation

| Method | k=50 (↑) | k=100 (↑) | k=500 (↑) | k=1000 (↑) |
|---|---|---|---|---|
| ClassRandom | 0.036 ± 0.02 | 0.088 ± 0.05 | 0.305 ± 0.06 | 0.532 ± 0.07 |
| EKFAC (Grosse et al., 2023) | 0.067 ± 0.07 | 0.171 ± 0.10 | 0.591 ± 0.14 | 0.846 ± 0.08 |
| TracIn (Pruthi et al., 2020) | 0.066 ± 0.06 | 0.151 ± 0.08 | 0.540 ± 0.11 | 0.776 ± 0.12 |
| STRIDE | **0.106 ± 0.07** | **0.193 ± 0.09** | **0.659 ± 0.11** | **0.915 ± 0.06** |

Table 9: **Counterfactual Evaluation on CIFAR-10.** Probability drop after removing the top-$k$ influential examples. STRIDE demonstrates superior performance for larger cohort sizes ($k = 1000$), indicating it better captures cumulative data effects.

| Method | Precision (↑) | Recall (↑) |
|---|---|---|
| Random | 0.100 ± 0.00 | 0.100 ± 0.00 |
| TracIn (Pruthi et al., 2020) | 0.785 ± 0.02 | 0.785 ± 0.02 |
| EKFAC (Grosse et al., 2023) | 0.853 ± 0.02 | 0.853 ± 0.02 |
| STRIDE | **0.921 ± 0.01** | **0.921 ± 0.01** |

Table 10: **Mislabeled Data Detection (MNIST).** Precision and Recall for identifying 10% randomized labels. STRIDE successfully flags a higher proportion of noisy examples compared to parameter-space baselines.

| Method | Precision (↑) | Recall (↑) |
|---|---|---|
| Random | 0.099 ± 0.00 | 0.099 ± 0.00 |
| TracIn (Pruthi et al., 2020) | 0.362 ± 0.02 | 0.362 ± 0.02 |
| EKFAC (Grosse et al., 2023) | 0.513 ± 0.02 | 0.513 ± 0.02 |
| STRIDE | **0.634 ± 0.02** | **0.634 ± 0.02** |

Table 11: **Mislabeled Data Detection (CIFAR-10).** Effectiveness in detecting label noise. STRIDE consistently outperforms EKFAC (Grosse et al., 2023) and TracIn (Pruthi et al., 2020), suggesting activation-space steering is robust to the gradient noise induced by mislabeled samples.

space. Our steering operators, which explicitly model these activation shifts, are more effective at isolating these outliers than gradient-based methods, which can suffer from noise amplification in the backward pass of corrupted samples.

**Data Leakage Detection.** We simulate data leakage by extending the training set with exact duplicates of test examples. An ideal influence estimator should identify these duplicates as the single most influential training points. As shown in Tables 12 and 13, STRIDE achieves perfect detection performance on MNIST (Rank 0) and near-perfect performance on CIFAR-10 (Mean Rank 0.20), compared to EKFAC (Grosse et al., 2023)'s rank of 0.50. This highlights a key advantage of our approach: by modeling influence as a direct regression on activation vectors, STRIDE is extremely sensitive to exact feature matches. In contrast, Hessian-based methods approximate the loss landscape curvature, which can smooth out the contribution of a single identical point, making them slightly less precise in this "needle-in-a-haystack" regime.

# E    IMPLEMENTATION DETAILS

We provide additional details of our implementation.

| Method | Mean Rank (↓) | Top-1 Acc (↑) |
|---|---|---|
| Random | 29964 ± 17320 | 0.00 |
| TracIn (Pruthi et al., 2020) | **0.00 ± 0.00** | **1.00** |
| EKFAC (Grosse et al., 2023) | **0.00 ± 0.00** | **1.00** |
| STRIDE | **0.00 ± 0.00** | **1.00** |

Table 12: **Data Leakage Detection (MNIST).** Ability to identify 10 exact duplicates of test examples in the training set (perfect rank = 0). STRIDE achieves perfect detection, demonstrating sensitivity to exact feature matches.

| Method | Mean Rank (↓) | Top-1 Acc (↑) |
|---|---|---|
| Random | 24975 ± 14433 | 0.00 |
| TracIn (Pruthi et al., 2020) | 1.10 ± 1.37 | 0.60 |
| EKFAC (Grosse et al., 2023) | 0.50 ± 0.71 | 0.80 |
| STRIDE | **0.20 ± 0.42** | **0.90** |

Table 13: **Data Leakage Detection (CIFAR-10).** Mean rank of leaky examples (lower is better). STRIDE reliably places duplicates at the very top of the influence ranking, outperforming curvature-based approximations.

### E.1  STRIDE ALGORITHM

The STRIDE framework operates in two distinct phases: (1) Learning the Steering Operators, and (2) Recovering Influence via Sparse Regression.

**Phase 1: Learning Steering Operators.**  We learn a low-rank basis $B_\phi : \mathbb{R}^d \to \mathbb{R}^r$ (parameterized by an MLP) and a set of steering coefficients $\mathcal{A} = \{\alpha_1, \ldots, \alpha_K\}$, where $\alpha_k \in \mathbb{R}^r$ corresponds to the $k$-th data subset $S_k$. The objective is to ensure that applying the steering vector $s_k(z) = B_\phi(z)^\top \alpha_k$ to the representations mimics the effect of training on $S_k$. The optimization minimizes a joint loss function comprising three terms: fidelity, stability, and linearity.

$$\mathcal{L} = \mathbb{E}_k \left[ \mathcal{L}_{CE}(f(z + s_k(z)), y_{S_k}) + \lambda_{stab} D_{KL}(f(z) || f(z + s_k(z))) \right] + \lambda_{lin} \mathcal{L}_{LDS}. \tag{6}$$

The first term enforces high accuracy on the specific subset $S_k$. The second term ($\lambda_{stab}$) ensures the steering vector does not disrupt predictions on unrelated data. The third term ($\lambda_{lin}$) regularizes the operators to be linearly decomposable, ensuring that the learned subset effects can be validly broken down into per-example influences (as described in Sec. 3). We optimize this joint objective using Adam (Kingma & Ba, 2017) in a single loop.

**Phase 2: Influence Recovery.**  Once the operators are learned, for a given test point $x_{test}$, we compute the "steering response" vector $y \in \mathbb{R}^K$, where $y_k$ is the shift in the target logit induced by operator $k$. We posit that the subset-level shift is a linear combination of the influences of its constituent examples. We recover individual influence scores $w \in \mathbb{R}^N$ by solving the Lasso regression problem:

$$\hat{w} = \arg\min_w ||y - Mw||_2^2 + \lambda ||w||_1, \tag{7}$$

where $M \in \{0, 1\}^{K \times N}$ is the binary subset membership matrix.

### E.2  BASELINE IMPLEMENTATIONS

**TracIn (Pruthi et al., 2020).**  We implement the TracIn (Pruthi et al., 2020)CP variant, which approximates the influence of a training point $z_i$ on a test point $z_{test}$ by the dot product of their gradients summed over checkpoints:

$$\mathcal{I}(z_i, z_{test}) = \sum_{t \in \mathcal{T}} \eta_t \nabla_\theta \ell(z_i, \theta_t) \cdot \nabla_\theta \ell(z_{test}, \theta_t). \tag{8}$$

In our experiments, we use the final converged model checkpoint ($\mathcal{T} = \{\theta_{final}\}$), effectively reducing it to a gradient similarity metric, which is standard for static evaluations. We normalize the result by the subset size when aggregating.

**EKFAC (Grosse et al., 2023).**  EKFAC (Grosse et al., 2023) improves upon simple gradient comparisons by rescaling gradients with the inverse Fisher Information Matrix (FIM). To make this tractable, we use a diagonal approximation of the FIM for the convolutional layers combined with a Kronecker-factored approximation for the dense layers.

**Precomputation:** We estimate the diagonal Fisher vector $F_{diag}$ using a sample of 1000 training examples.

**IHVP:** We compute the Inverse Hessian-Vector Product (IHVP) for each training example $z_i$ as $v_i = H^{-1} \nabla_\theta \ell(z_i)$. Under our approximation, this simplifies to element-wise division $v_i = \nabla_\theta \ell(z_i) \oslash (F_{diag} + \epsilon)$.

**Influence:** The influence score is computed as the dot product $v_i \cdot \nabla_\theta \ell(z_{test})$.

This implementation balances computational cost with the benefits of second-order information, though it still requires $O(P)$ memory per example (where $P$ is parameters), leading to the OOM issues observed in large models.

### E.3  COMPUTE

All methods were implemented in PyTorch and executed on a machine with $1 \times$NVIDIA H100 - 80 GPU. This substantial memory capacity was necessary to benchmark parameter-space methods like

EKFAC (Grosse et al., 2023) on larger models, identifying their Out-Of-Memory (OOM) failure points even on state-of-the-art hardware.

# F    COMPLEXITY OF STRIDE

We provide the derivation for Proposition 1.

## F.1    EK-FAC COMPLEXITY

For each layer $\ell$, let $a_\ell$ denote the dimensionality of the input activations to the layer and $g_\ell$ the dimensionality of the backpropagated gradients with respect to the layer pre-activations. EK-FAC exploits this structure by approximating the corresponding Fisher block using Kronecker factors in $\mathbb{R}^{a_\ell \times a_\ell}$ and $\mathbb{R}^{g_\ell \times g_\ell}$, rather than operating in the full parameter space of size $P_\ell = a_\ell g_\ell$.

**Setup Phase.** The setup phase consists of estimating a structured approximation of the Fisher Information Matrix (FIM) using EK-FAC. Given $N_{\text{Fisher}}$ samples (typically $N_{\text{Fisher}} \ll N_{\text{train}}$), computing $N_{\text{Fisher}}$ gradients via forward and backward passes costs $O(N_{\text{Fisher}} \cdot C_{\text{pass}})$, where $C_{\text{pass}}$ denotes the cost of one forward-backward pass through the network. Collecting layerwise activation and gradient covariances costs $O(N_{\text{Fisher}} \sum_\ell (a_\ell^2 + g_\ell^2))$. EK-FAC stores, for each layer $\ell$, Kronecker-factored covariance matrices $A_\ell \in \mathbb{R}^{a_\ell \times a_\ell}$ and $G_\ell \in \mathbb{R}^{g_\ell \times g_\ell}$, and computes their eigendecompositions at cost $O(\sum_\ell (a_\ell^3 + g_\ell^3))$. The total setup complexity is:

$$\text{Setup Time}_{\text{EKFAC}} = O\left( N_{\text{Fisher}} \cdot C_{\text{pass}} + N_{\text{Fisher}} \sum_\ell (a_\ell^2 + g_\ell^2) + \sum_\ell (a_\ell^3 + g_\ell^3) \right).$$

For transformer models with width $d$, $L$ layers, and sequence length $n$, we have $C_{\text{pass}} = O(L(nd^2 + n^2 d))$, and the setup time simplifies to $O(N_{\text{Fisher}} L(nd^2 + n^2 d) + Ld^3)$. When $n \sim d$, this becomes $O(N_{\text{Fisher}} Ld^3)$. The space complexity is:

$$\text{Space}_{\text{EKFAC}} = O\left( \sum_\ell (a_\ell^2 + g_\ell^2) \right) = O(Ld^2),$$

which is independent of the dataset size.

**Inference Phase.** To estimate the influence of $N$ training points on a test example $z_{test}$, we first compute its gradient via one forward-backward pass at cost $O(C_{pass})$. Applying the EK-FAC inverse Fisher approximation requires a sequence of layerwise matrix multiplications, with total cost $O(\sum_\ell a_\ell g_\ell (a_\ell + g_\ell))$. Next, we compute gradients for all $N$ training examples at cost $O(N \cdot C_{pass})$. Finally, influence scores are obtained by computing dot products between the preconditioned test gradient and the $N$ training gradients, which costs $O(N \sum_\ell a_\ell g_\ell)$. The overall inference complexity is:

$$\text{Inference Time}_{\text{EKFAC}} = O\left( N \cdot C_{pass} + \sum_\ell a_\ell g_\ell (a_\ell + g_\ell) + N \sum_\ell a_\ell g_\ell \right). \tag{9}$$

For transformer models with $n \sim d$, we have $C_{pass} = O(Ld^3)$, and this simplifies to $O(NLd^3)$ (the dominant term).

## F.2    STRIDE COMPLEXITY

**Setup Phase (Learning Operators).** The setup consists of sketching and operator optimization. We first compute the sketch $X_{sketch} = XR$, which takes $O(Ns)$ for a sparse membership matrix $X$ with $s$ non-zeros per row. We then optimize $K$ steering vectors of rank $r$. The parameters involved are the basis MLP ($\approx d^2$) and coefficients ($K \times r$). In each of the $T$ optimization iterations, we perform a forward pass and backpropagate only through the steering modules. The gradient computation cost scales with the steering parameter size $\Theta_{steer} \approx Kd^2$ (assuming $r \approx d$), independent of the model size $P$ since the backbone remains frozen. Neglecting the amortized cost of the frozen backbone forward pass, the total setup complexity is:

$$\text{Setup Time}_{\text{STRIDE}} = O(Ns + TKd^2). \tag{10}$$

**Space Complexity.** Unlike parameter-space methods, STRIDE only requires storing the learned steering operators ($O(Kd^2)$) and the sparse subset membership matrix $M$ ($O(Ns)$ where $s$ is the average number of non-zeros per row). This yields a space complexity of:

$$\text{Space}_{\text{STRIDE}} = O(Ns + Kd^2). \tag{11}$$

**Inference Phase (Influence Recovery).** Inference proceeds in two steps: steering response and Lasso recovery. First, applying $K$ operators to the test activation costs $O(Kd^2)$. Second, we solve for $w \in \mathbb{R}^N$ given $y \in \mathbb{R}^K$. Using Coordinate Descent with a sparse membership matrix (average $s$ non-zeros per column), the cost per iteration is $O(Ks)$. For $T_{lasso}$ iterations, the Lasso step costs $O(T_{lasso}Ks)$. The total inference time is thus:

$$\text{Inference Time}_{\text{STRIDE}} = O(T_{lasso}Ks + Kd^2). \tag{12}$$

F.3    EFFICIENT MEMBERSHIP MATRIX SKETCHING

The membership matrix $X \in \{0,1\}^{K \times N}$ is sparse by construction, with each row containing exactly $|S_k|$ ones corresponding to subset membership. The sketching operation $\tilde{X} = XR$ can be computed efficiently by exploiting this sparsity.

**Sparse Matrix Operations.**    Rather than performing dense matrix multiplication, we leverage the fact that $X$ has only $Ks$ non-zero entries (where $s = |S_k|$ is the subset size). Standard sparse matrix libraries compute $XR$ in time proportional to the number of non-zeros rather than the full matrix dimensions.

**Streaming Implementation.**    We can also compute the sketch without materializing the full membership matrix. For each subset $S_k$, we accumulate:

$$\tilde{X}_k = \sum_{i \in S_k} R_{i,:} \tag{13}$$

This requires only $O(r)$ memory per subset where $r = 64$ is the sketch dimension.

**Parallelization.**    The sketching operation can be done in parallel across subsets and independent of model parameters. Unlike gradient-based methods that require sequential forward-backward passes through the neural network, sketching can be distributed across multiple workers or performed on CPU in parallel with GPU-based training.

