# OpenReview forum: "STRIDE: Training Data Attribution Can Be Estimated In Activation Space"
_ICLR.cc/2026/Workshop/Sci4DL — Sci4DL 2026_

### Official Review · Reviewer_kPF7 · 2026-02-24

**Fit:** 3
**Significance:** 2
**Confidence:** 2

**Summary:**

The authors introduce a method for training data attribution (TDA), STRIDE, which avoids parameter count scaling issues by operating solely in activation space. This works by learning low-rank steering operators that approximate the effect of re-training on subsets of the training data (e.g., image classes).

**Strengths:**

- Operating solely in activation space is attractive, as scaling to larger parameter counts is a known limitation of TDA methods. The methodology, loss functions etc. seem suitable, at least for the evaluated task of classification. Overall, I think this shows promise (though evaluation in more realistic scenarios is needed), and I'm quite curious to see whether the promise of avoiding parameter-count scaling makes this suitable for LLMs in practice.
- Suitable evaluation at the workshop stage, although MNIST and CIFAR-10 are somewhat toy datasets.
- The paper is well-written and organized, with extensive and helpful supplementary materials.

**Suggestions:**

Extending to LLMs is a necessary next step to demonstrate the practicality of this work. However, the work would benefit from a discussion of how this could be implemented in practice and the potential technical challenges. For example, I'd like to see the authors discuss:
- STRIDE is formulated for classification tasks. How would this extend to LLMs, e.g. by simply applying it to predicted tokens by the LM head? If that's the case, you're now dealing with many more classes over the entire vocabulary; would STRIDE scale well with this? Or, is there a more principled way to organize data into subsets in this domain?
- Also, in this setting, do the authors have evidence that the key assumption for STRIDE that "removing a small subset induces an approximately additive, locally linear perturbation in activation space" will apply? If it doesn't, would the method just take some performance hits, or fail altogether?
- Although scaling up to LLMs seems to be the main interest, how might STRIDE scale to larger vision models, even just imagenet classifiers? MNIST and CIFAR-10 are basically toy datasets at this point; do the authors expect performance gains to be similar for ImageNet models, considering for example that ImageNet has many more classes, which are also significantly more finely separated as compared to MNIST and CIFAR-10? Would this particular latter issue make the steering operator learning more challenging?

---

### Official Review · Reviewer_SLRi · 2026-02-27

**Fit:** 2
**Significance:** 1
**Confidence:** 2

**Summary:**

The paper proposes STRIDE,  a two-stage framework for training data attribution that avoid parameter-space influence computations by working in activation space. In stage 1, STRIDE learns subset-specific low-rank steering operators intended to approximate the logit that would occur under removing a subset. In stage 2, it regresses subset-level effects back to per-example scores using Lasso regression. The experiments are on MNIST and CIFAR-10 with large efficiency gains in inference time.

**Strengths:**

- A clean overview figure that makes the two-stage pipeline easy to follow.
- The related-work section is clearly structured and easy to navigate.
- STRIDE is orders of magnitude faster at inference and supports straightforward batching since it operates on activations rather than gradient.

**Suggestions:**

**Clarity:**

- Since this is a workshop venue, I mainly evaluate novelty and technical soundness. However, if the authors plan to submit to a flagship conference, the main text should provide concise definitions of $\mathcal{L}_{\text{fid}}$, $\mathcal{L}_{\text{stab}}$, and $\mathcal{L}_{\text{lin}}$ (with a brief intuition for each term.

- It is unclear why the method needs to solve for $\omega$ instead of directly using $Y_z$ as the influence signal. As written, $Y_z$ appears to be a subset-level effect vector, while $\omega$ is the recovered per-example influence. This distinction (and the role of overlapping vs. disjoint subsets, if applicable) should be clarified explicitly.

**Evaluation:**

- The evaluation is somewhat restricted: only MNIST and CIFAR-10 are included, and the gains appear marginal. It would strengthen the paper to include a larger-scale dataset (e.g., Tiny ImageNet), especially given the reported access to strong compute.

---

### Official Review · Reviewer_Q7QJ · 2026-02-27

**Fit:** 1
**Significance:** 2
**Confidence:** 2

**Summary:**

The paper introduces STRIDE, a scalable and effective method for data attribution.

STRIDE splits the training data into random subsets and extracts low-rank encodings from a target model with frozen weights such that, when multiplied by subset-specific weights, they can recover the model’s predictions. At test time, for an input \(x\), the method computes a sparse vector representing the influence of training samples on the prediction for \(x\). This relies on the assumption that the per-subset representations of \(x\), obtained by multiplying its encodings with the subset-specific weights, can be linearly reconstructed from training sample influences.

STRIDE outperforms competing methods across several benchmarks and applications, including counterfactual evaluation, mislabeled data detection, and data leakage detection. It also demonstrates a substantial advantage over competing approaches in both efficiency and test-time performance, at the cost of a modest increase in setup time.

The hyperparameters of STRIDE (in particular the rank and subset size) allow practitioners to manage the trade-off between available RAM and attribution effectiveness. I recommend including additional experiments that more systematically explore this trade-off.

**Strengths:**

The paper provides a comprehensive study. The proposed method demonstrates its effectiveness across various benchmarks and applications. It also shows a significant advantage over competing methods in terms of efficiency and test-time performance, at the cost of slightly higher setup time, which can be negligible when many samples are evaluated at test time.

The fact that the hyperparameters can be adjusted to avoid out-of-memory issues and enable running the method on a single GPU is also an important and practical feature of STRIDE.

**Suggestions:**

- Shouldn’t the fidelity loss be defined with respect to the model’s predicted labels (c) rather than the ground-truth labels (y)?
- I am curious about how the steering rank affects the final results.

---

### Meta-Review · Area_Chair_HNFa · 2026-02-28

**Recommendation:** Accept

**Metareview:**

This papers introduces STRIDE, a data-attribution method in activation space rather than parameter space. While evaluation is limited, all reviewers noted that the results are promising. I recommend acceptance.

---

### Decision · Program_Chairs · 2026-03-02

Accept